# Rutin Attenuates H_2_O_2_-Mediated Oxidative Stress, Inflammation, Endoplasmic Reticulum Stress and Apoptosis in Bovine Mammary Epithelial Cells via the AMPK/NFE2L2 Signaling Pathway

**DOI:** 10.3390/ijms26104788

**Published:** 2025-05-16

**Authors:** Hongyan Ding, Weizhe Yan, Daoliang Zhang, Lei Wang, Yue Yang, Chang Zhao, Shibin Feng, Xichun Wang, Jishun Tang, Dong Wu, Jinjie Wu, Yu Li

**Affiliations:** 1College of Veterinary Medicine, Anhui Agricultural University, Hefei 230036, China; dinghy1988@163.com (H.D.); wolf1368755106@163.com (W.Y.); zdl15936569251@163.com (D.Z.); leiwang233815@163.com (L.W.); yanghappy2085@163.com (Y.Y.); chang_zhao@ahau.edu.cn (C.Z.); fengshibin@ahau.edu.cn (S.F.); wangxichun@ahau.edu.cn (X.W.); wjj@ahau.edu.cn (J.W.); 2Anhui Province Key Laboratory of Livestock and Poultry Product Safety Engineering, Institute of Animal Science and Veterinary Medicine, Anhui Academy of Agricultural Sciences, Hefei 230001, China; tjs157@163.com (J.T.); wud1971@126.com (D.W.)

**Keywords:** rutin, oxidative stress, inflammatory, apoptosis, endoplasmic reticulum stress, AMPK/NFE2L2 signaling pathway

## Abstract

Transition dairy cows face severe oxidative stress that disrupts mammary epithelial homeostasis through intertwined oxidative, inflammatory, and endoplasmic reticulum (ER) stress pathways. This study hypothesized that rutin, a natural flavonoid, alleviates hydrogen peroxide (H_2_O_2_)-induced oxidative damage in bovine mammary epithelial cells (BMECs) via AMPK/NFE2L2 signaling activation. In this study, BMECs were pre-incubated with rutin. Subsequently, cells were treated with or without H_2_O_2_. Additionally, by transfecting BMECs with NFE2L2 siRNA (siNFE2L2), we investigated how AMPK/NFE2L2 signaling mediated by rutin may prevent H_2_O_2_-induced oxidative damage. The results show that increases in reactive oxygen species (ROS), expression of inflammatory cytokines, expression of proteins related to endoplasmic reticulum stress and the apoptosis rate induced by H_2_O_2_ in cells, were attenuated in rutin cultures. Challenges with H_2_O_2_ led to a lower abundance of proteins related to AMPK and NFE2L2. Comparatively, these effects were reversed in cultures with rutin. Transfection with siNFE2L2 reversed the protection of rutin, suggesting that NFE2L2 is essential for the protective mechanism of rutin. These results elucidated the molecular mechanism of rutin’s resistance to H_2_O_2_-mediated oxidative injury through the AMPK/NFE2L2 signaling pathway and suggested that it could be used as a potent in vivo antioxidant for ruminants during periods of stress, such as before and after calving.

## 1. Introduction

As mitochondrial electron transport chain by-products, reactive oxygen species (ROS) play central roles in regulating numerous physiological and biological responses [1]. Under physiological conditions, ROS levels are maintained in dynamic equilibrium through the modulation of cellular processes that both produce ROS as a byproduct of aerobic metabolism and neutralize/eliminate them by cellular antioxidative mechanisms [2]. In other words, oxidative stress results from redox imbalance. Excessive ROS or referred to as oxidative stress is able to induce damage to cells and tissues [3]. In high-yielding dairy cows, this phenomenon is exacerbated during the transition period (3 weeks prepartum to 3 weeks postpartum) due to drastic metabolic shifts. The onset of lactation triggers a three- to five-fold increase in hepatic oxygen consumption, while concurrent negative energy balance promotes excessive lipolysis and mitochondrial–oxidation dual pathways that synergistically amplify ROS generation [4,5]. In the past ten years, research has consistently shown that high-yielding and transition dairy cows frequently undergo oxidative stress, increasing their vulnerability to various diseases [4,5,6,7]. Notably, over 60% of postpartum disorders (e.g., mastitis, metritis, and retained placenta) are epidemiologically associated with systemic oxidative stress markers, underscoring their role as a keystone pathological driver in modern intensive dairy systems.

Belonging to heterotrimeric serine/threonine kinases, AMP-activated protein kinase (AMPK) is a crucial player in response to various metabolic stresses in cells [8]. Nuclear factor erythroid 2-related factor 2 (NFE2L2, alternatively referred to Nrf2) serves as a downstream transcription factor of AMPK, and which is known to affect antioxidant response as the ‘master regulator’ [9]. In a study of BMEC in vitro, compared with the control, the treatment of cells with 600 μM of H_2_O_2_ for 6 h sharply decreased the cell proliferation rate and SOD and GSH-Px activities and strongly enhanced ROS and MDA production and caspase-3 activity [10]. H_2_O_2_ was also shown to promote the production of ROS and MDA to generate higher rates of apoptosis and oxidative stress in BMEC. Compared with the control, cells transfected with NFE2L2-siRNA3 with or without H_2_O_2_ had lower production of ROS and MDA, as well as activities of SOD, CAT, GSH-Px, and GST. In contrast, BMECs transfected with pCMV6-XL5-NFE2L2 plasmid, which upregulates the expression of NFE2L2, had greater proliferation rates and lower apoptosis rates, while the production of ROS and MDA increased markedly [11]. These results highlight the critical function of NFE2L2 in controlling oxidative stress levels in BMECs.

Rutin is a common dietary flavanol, abundantly found in plants, which is very popular for its antioxidant properties [12]. In humans and rats, numerous studies have shown that rutin may significantly decrease intracellular oxidation, increase antioxidant enzyme activity and inhibit apoptosis [13,14,15]. Of relevance to antioxidant intervention strategies, Stoldt et al. demonstrated that dietary rutin supplementation (100 mg/kg BW for 14 days) in transition dairy cows not only elevated plasma glucose and albumin—indicators of improved hepatic gluconeogenesis and protein synthesis—but also paradoxically increased β-hydroxybutyrate (BHB) levels [16]. This metabolic profile suggests that rutin may enhance antioxidant capacity while modulating energy substrate partitioning, potentially through redox-mediated regulation of hepatic mitochondrial function [16]. Notably, mechanistic studies in bovine mammary epithelial cells (BMECs) have established H₂O₂-induced oxidative stress as a validated experimental model. For instance, lycopene was shown to alleviate H_2_O_2_-induced ROS accumulation in BMECs through NFE2L2 pathway activation, simultaneously suppressing inflammatory cytokines and apoptosis [17]. This model’s effectiveness in replicating redox imbalance makes it particularly suitable for investigating rutin’s potential mechanisms. However, the precise mechanism underlying rutin supplementation in dairy cows remains uncertain.

BMECs not only participate in milk production but they are also essential participants in the first line of defense against the invasion of pathogens [18]. Therefore, our general hypothesis is that rutin modulates the proinflammatory and antioxidant responses of mammary cells during oxidant stress simulation. The present study aims to assess whether rutin supplementation modulates H_2_O_2_-triggered immune and antioxidant responses in BMECs in vitro.

## 2. Results

### 2.1. Impacts of H_2_O_2_ and Rutin on Cell Viability

This study used the CCK-8 assay to investigate the effects of rutin on the viability and H_2_O_2_-induced damage of BMECs. After treatment with different concentrations of H_2_O_2,_ BMEC viability decreased dose-dependently (Figure 1A). Based on previous research reports and the results of this study, cells stimulated by 600 μmol/L H_2_O_2_ for 6 h were selected as the experimental conditions in further experiments [11]. Rutin applied at varying concentrations (0, 10, 25, 50, and 100 μmol/L) showed no toxic effect on BMEC for 1, 3, 6, 12, and 24 h (Figure 1B). Additionally, the cell viability was higher following 12 h of treatment with 25 μmol/L rutin (*p* < 0.05), and relatively higher with 10 or 50 μmol/L rutin for 12 h (*p* > 0.05, Figure 1B). Therefore, 12 h of stimulation by rutin was chosen for later experiments. As shown in Figure 1C, treatment with 600 μmol/L H_2_O_2_ significantly attenuated BMEC viability compared with the control group (*p* < 0.05), but rutin restored the cell viability weakened by H_2_O_2_ treatment and the effect was the most enhanced in the group pretreated with 50 μmol/L rutin.

### 2.2. Effect of H_2_O_2_ and Rutin on ROS in BMEC

The intracellular ROS level was greater with 600 μmol/L H_2_O_2_ compared with the control group (Figure 2). The ROS level decreased with 25, 50, and 100 μmol/L rutin compared with that in the H_2_O_2_ group (*p* < 0.05) and the lowest was 50 μmol/L rutin.

### 2.3. Rutin Mitigated Oxidative Stress Induced by H_2_O_2_

As indicated in Figure 3, a higher MDA content plus lower activities of SOD, GSH-Px and CAT and T-AOC content were observed in the treatment with 600 μmol/L H_2_O_2_ compared with the control group (*p* < 0.05). Moreover, the MDA content dropped whereas the activities of CAT rose after treatment with 25, 50, and 100 μmol/L rutin compared with the H_2_O_2_ group (*p* < 0.05). Treatment with 10, 25, 50, and 100 μmol/L rutin led to increases in SOD and GSH-Px activities and T-AOC content (*p* < 0.05), with 50 μmol/L rutin resulting in the highest increase.

### 2.4. Rutin Caused Activation of the AMPK/NFE2L2 Pathway

H_2_O_2_ could significantly decrease the phosphorylation of AMPKα protein and nuclear NFE2L2 protein levels compared to the control group (*p* < 0.05, Figure 4). Rutin, in contrast with H_2_O_2_, increased phosphorylation of AMPKα as well as nuclear NFE2L2 protein levels (Figure 4A–C). In addition, rutin significantly increased the expression *NFE2L2*, *NQO1* and *HMOX1* at the mRNA expression level (Figure 4D–F).

### 2.5. Rutin Rescued the Activity of the NF-kB Signaling Pathway Elicited by H_2_O_2_

The p-p65/p65 and p-IκBα/IκBα ratios were upregulated significantly by H_2_O_2_ (Figure 5A–C). Such ratios decreased in the rutin group in comparison to those in the H_2_O_2_ group. Similarly, the H_2_O_2_ group exhibited upregulated *TNF-α*, *IL-1β* and *IL-6* in terms of the mRNA expression, while rutin attenuated the increase in the inflammatory cytokine mRNA expression induced by H_2_O_2_ (Figure 5D–F).

### 2.6. Rutin Prevented H_2_O_2_-Mediated Caspase Pathway Activation and Apoptosis

As shown in Figure 6, 600 μmol/L H_2_O_2_ increased Bax, Caspase3, and Caspase9 protein expression and decreased Bcl-2 expression in comparison with the control group (*p* < 0.05). Moreover, compared to H_2_O_2_, rutin decreased the expression of Bax, Caspase3, and Caspase9 but increased Bcl-2 expression (*p* < 0.05). Results showed that compared with the control group, the Bax/Bcl-2 ratio in H_2_O_2_-treated BMECs significantly increased from 0.3951 to 1.7694, suggesting that pro-apoptotic signaling was markedly enhanced. With increasing concentrations of rutin, the Bax/Bcl-2 ratio gradually decreased to 1.4592 (10 μmol/L), 1.1620 (25 μmol/L), and 0.7381 (100 μmol/L), respectively. Notably, the ratios in the 25 μmol/L and 100 μmol/L groups were significantly lower than those in the H_2_O_2_ group (*p* < 0.05), indicating that rutin at certain concentrations can inhibit oxidative stress-induced apoptosis in BMECs. The ratio in the 100 μmol/L group approached that of the control group (0.7381 vs. 0.3951), suggesting its strong protective effect against oxidative damage.

### 2.7. Rutin Attenuated H_2_O_2_-Induced Endoplasmic Reticulum Stress (ERS)

Compared with the control group (0 μmol/L rutin + 0 μmol/L H_2_O_2_), 600 μmol/L H_2_O_2_ increased the expression of the GRP78 and CHOP proteins (*p* < 0.05) along with the mRNA expression of *GRP78* and *ATF4* (*p* < 0.05, Figure 7). Compared with the H_2_O_2_ group, treatment with rutin and H_2_O_2_ reduced the mRNA expression of *GRP78* and *ATF4* in addition to GRP78 and CHOP protein expression.

### 2.8. Rutin Upregulated the NFE2L2 Signaling Pathway in BMEC Through AMPK

As shown in Figure 8, rutin increased AMPK phosphorylation in comparison to the control group. After adding an AMPK inhibitor, the rutin-induced phosphorylation of AMPK was blocked (Figure 8A,B). Interestingly, the nuclear translocation of the NFE2L2 protein was inhibited and its mRNA expression was decreased by the addition of the AMPK inhibitor to BMECs (Figure 8C,D).

### 2.9. Rutin Exploited NFE2L2 to Alleviate Oxidative Stress, Inflammatory Responses, Apoptosis, and ERS Induced by H_2_O_2_

For the purpose of determining NFE2L2’s role in rutin’s protection against H_2_O_2_-mediated oxidative damage to BMEC, NFE2L2 expression was repressed using NFE2L2 siRNA. As shown in Figure 9, the transfection with NFE2L2 siRNA significantly reduced NFE2L2 at both mRNA and protein expression levels in contrast to the control group and negative control group (NC). ROS generation was decreased by rutin–H_2_O_2_ treatment comparison to that in the H_2_O_2_ group. On the contrary, ROS generation in the siNFE2L2–rutin–H_2_O_2_ group was significantly enhanced in comparison with that in the rutin–H_2_O_2_ group (Figure 10A). In contrast to the H_2_O_2_ group, the rutin–H_2_O_2_ treatment significantly repressed inflammatory cytokines (*TNF-α*, *IL-1β*, and *IL-6*; Figure 10B–D) and genes related to ERS (*GRP78*, *ATF4*, and *CHOP*; Figure 10F,G) at the mRNA expression lever. *TNF-α*, *IL-1β*, *IL-6*, *GRP78*, *ATF4*, and *CHOP* mRNA expressions were increased by siNFE2L2–rutin–H_2_O_2_ treatment compared with that with rutin–H_2_O_2_. Furthermore, compared with the H_2_O_2_ group, the rutin–H_2_O_2_ treatment decreased the apoptosis rate. This coincides with the changes in inflammatory cytokines, ROS and abundance of ERS-related genes. In contrast to rutin–H_2_O_2_, siNFE2L2–rutin–H_2_O_2_ treatment elevated the apoptosis rate (Figure 10E).

## 3. Discussion

The periparturient period imposes severe metabolic and oxidative stresses on bovine mammary glands, particularly in cows with subclinical/clinical ketosis [19,20]. As oxidative stress intersects with inflammation, apoptosis, and endoplasmic reticulum stress (ERS) [21], enhancing mammary antioxidant capacity emerges as a strategic intervention to safeguard tissue homeostasis and lactation performance.

Our findings establish rutin—a bioactive flavonoid with multimodal cytoprotective properties [22]—as a potent mitigator of H_2_O_2_-induced oxidative damage in bovine mammary epithelial cells (BMECs). Rutin directly scavenges ROS while upregulating endogenous antioxidants (CAT, GSH-Px, SOD, T-AOC), corroborating its dual antioxidant mechanism observed across species [23]. Crucially, this redox modulation underpins rutin’s broader protective effects.

By blocking H_2_O_2_-triggered NF-κB activation, rutin suppresses proinflammatory cytokines (TNF-α, IL-6, IL-1β) in BMECs—a pattern aligning with its inhibition of NF-κB/MAPK pathways in muscle cells [24] and endothelial models [25]. This conserved anti-inflammatory response highlights rutin’s potential to disrupt oxidative-inflammatory crosstalk, a key driver of mammary pathophysiology [26].

Rutin has a dual effect on apoptosis. Rutin promotes cell apoptosis in cancers such as pancreatic carcinoma (PANC-1), colon cancer (HT29 and Caco-2) and lung cancer (A549) [27,28]. Rutin pretreatment alleviates the cardiomyocytes apoptosis induced by sepsis [29]. Rutin blocks the activity of Caspase3, Caspase8, and Caspase9 in triethylene glycol dimethacrylate-stimulated RAW264.7 macrophages [30]. In terms of the protein expressions in this study, rutin reduced Bax, Caspase3, and Caspase9, while Bcl-2 was increased. The aforementioned results indicated that rutin attenuated H_2_O_2_-induced apoptosis in BMEC.

ERS refers to a cellular process resulting from serious diversified stress conditions such as oxidative stress [31]. GRP78 and CHOP were used as markers for ERS [32]. In a study of H9C2 cells treated with high glucose, rutin was capable of suppressing GRP78, ATF6, and CHOP expression in a dose-dependent manner [33]. In mice, rutin protects against toxicity in the liver and kidney by inhibiting the expression of ATF6, GRP78, and CHOP induced by malathion [34]. In the present study, rutin was confirmed to repress the expression of ERS-related protein induced by H_2_O_2_ in BMECs.

AMPK is not only a major player in modulating cellular energy balance but also an upstream protein of NFE2L2 [35]. Molecular pathways associated with several flavonoids and other phenolic substances involve the AMPK/NFE2L2 pathway. Sophoricoside attenuates LPS-mediated acute lung injury in a manner dependent on AMPK/NFE2L2 [36]. In both in vivo and in vitro studies, niacin can activate the AMPK/NFE2L2 signaling pathway to alleviate mastitis in dairy cows [37]. Lentinan activates the NFE2L2 signaling pathway to relieve LPS-triggered inflammation plus apoptosis in BMECs while repressing oxidative stress [38]. Meloxicam inhibited the inflammatory response, together with oxidative stress in bovine endometrial epithelial cells treated by LPS via the NFE2L2 and NF-κB pathways [39]. Rutin administered to mice reduced oxidative stress to mitigate diabetic neuropathy by means of HMOX1 and NFE2L2 [40]. In this study, rutin activated AMPK/NFE2L2 signaling pathway. However, the knockdown of NFE2L2 by siRNA significantly reduced the improvement effect of rutin on oxidative damage of BMEC. In summary, rutin reduces hydrogen peroxide-induced oxidative stress, inflammation, apoptosis and ERS in BMECs by activating NFE2L2.

By concurrently targeting oxidative stress, inflammation, apoptosis, and ERS through a unified signaling mechanism, rutin offers a phytoceutical strategy to enhance mammary resilience during metabolic crises. This cost-effective approach could complement current management practices, reducing reliance on antibiotics while supporting animal welfare and productivity.

## 4. Materials and Methods

### 4.1. Culture and Treatment of Cells

The bovine mammary epithelial cells (BMEC) purchased from Ningbo Mingzhou Biotechnology Co., Ltd. (MZ-2690, Ningbo, China) were cultured in a humidified incubator containing 5% carbon dioxide at 37 °C. BMEC were cultured in complete medium (RPMI1640 (Gibco, Grand Island, NY, USA, C11875500BT) + supplemented with 10% heat-inactivated fetal bovine serum (FBS; Gibco, Grand Island, NY, USA, 10270106), 100 U/mL penicillin and 100 μg/mL streptomycin (Beyotime, Shanghai, China, C0222). Six-well plates were applied to implant the cells (2 × 10^5^ in total) for 6 h of incubation with 600 μmol/L H_2_O_2_ or 12 h of preincubation with rutin at varying concentrations (0, 10, 25, 50 and 100 μmol/L). The concentration and duration of H_2_O_2_ treatment were chosen according to a study by Ma et al. [11]. Cell viability was confirmed >95% by CCK-8.

### 4.2. Assay on Cell Viability

A CCK-8 kit (APExBIO Technology LLC, Houston, TX, USA) was employed to appraise cell viability as per the manufacturer’s protocols. In brief, culture plates (96-well) were utilized for 24 h cell plating at an optimal density (1 × 10^5^ cells/mL) and then treated for 6 h using H_2_O_2_ (0, 200, 400, 600, 800, 1000 μmol/L) or 1, 3, 6, 12, 24 h with rutin (0, 10, 25, 50, 100 μmol/L). Subsequently, blended with CCK-8 solution (10 μL/well) was added to the cells for 37 °C incubation for 2 h, and a microplate reader was applied to monitor the absorbance at 450 nm (Thermo FisherScientific, Grand Island, NY, USA). The control group, without exposure to either rutin or H_2_O_2_, was utilized to obtain the data for normalization from each treatment group. The experiments included 5 replicates per treatment and were repeated at least twice.

### 4.3. Small Interference RNA Infection

The siRNA primer sequence targeting the NFE2L2 coding region involved sense, 5′-CTGGAGCAAGATTTAGATCAT-3′, plus antisense, 5′-ATGATCTAAATCTTGCTCCAG-3′, which were designed by reference to a previous study [11] and synthesized by General Biosystems Anhui Co., Ltd. (Chuzhou, China). The control siRNA (NC) or siNFE2L2 was transfected into BMECs using Lipofectamine 2000 reagent (Invitrogen, Carlsbad, CA, USA). Western blots and qPCR were used to evaluate the efficiency of the siNFE2L2.

### 4.4. Determination of ROS Level

A ROS detection kit (Elabscience Biotechnology Co., Wuhan, China) was used to measure intracellular ROS following the manufacturer’s protocols. After treatment, as described in cell culture and treatment, 10 μmol/L DCFH-DA was added, and cells were incubated for 20 min. Cells were then washed and re-suspended in reagent working solution 3 provided by the assay. Flow cytometry (BD Biosciences, Franklin Lakes, NJ, USA) was conducted to determine fluorescence. Intracellular ROS production was analyzed by fluorescence and normalized to the control. Three replicates were included for each treatment and experiments were repeated at least twice.

### 4.5. Oxidative Stress Assay

Levels and activities of oxidative indicators (SOD, T-AOC, CAT, MDA and GSH-Px) were used to evaluate oxidative stress using respective spectrophotometric diagnostic kits from Shanghai Enzyme-linked Biotechnology Co., Ltd. (Shanghai, China) following the protocols supplied by the manufacturer.

### 4.6. Cell Apoptosis Determination

An annexin V-FITC/propidium iodide apoptosis detection kit (Beyotime Biotechnology Co., Ltd., Shanghai, China) was selected for flow cytometry to measure cell apoptosis following the manufacturer’s protocol. In brief, the BMECs were treated with rutin and H_2_O_2_ as above. After collecting and washing twice in PBS, cells underwent resuspension in binding buffer (200 μL). Cell suspensions were then supplemented with Annexin V-FITC to a volume of 5 μL, followed by room-temperature incubation (10 min) away from light. Subsequently, 10 μL of propidium iodide was added to the cell suspension. The level of cell apoptosis was measured immediately.

### 4.7. RNA Isolation Plus Real-Time Quantitative PCR (qPCR)

These procedures were conducted by methods reported previously. Briefly, the total RNA of BMECs was extracted using Trizol (Takara Biotechnology Co., Ltd., Dalian, China). Then, a PrimeScript™RT reagent Kit with gDNA Eraser (TaKaRa Biotechnology Co., Ltd., Tokyo, Japan) was used for reverse transcriptiossn of RNA (1 μg in total) into cDNA in line with the manufacturer’s instructions. The reaction lasted for 15 min at 37 °C and 5 s at 85 °C. The selected primers are shown in Table 1. The mRNA abundance was detected using Novostart SYBR qPCR SuperMix Plus (Novoprotein, Nanjing, China) and a 7500 Real-Time PCR System (Applied Biosystems, Carlsbad, CA, USA). The reaction conditions were as follows: 3 min of 95 °C pre-denaturation, followed by 20 s of 95 °C denaturation and 1 min of 60 °C annealing for 40 cycles. The 2^−ΔΔCT^ method was used for the computation of relative expression, with *β-actin* expression for normalization.

### 4.8. Protein Extraction and Western Blotting

Protein extraction and Western blotting were implemented using methods described previously [17]. Specifically, cell lysis buffer and a nuclear extraction kit (Beyotime, Shanghai, China) were used to extract total proteins and nuclear proteins, respectively. Following SDS-PAGE for fractionation, the proteins were transferred onto PVDF membranes. Subsequently, primary antibodies were added onto the membranes for overnight incubation at 4 °C, after which HRP-conjugated secondary antibodies (SA00001-2, 1:5000, Proteintech, Rosemont, IL, USA) was added. The primary antibodies used included anti-NFE2L2 (ab137550, 1:1000), anti-Caspase3 (ab32351, 1:500), anti-Caspase9 (ab32539, 1:500) and anti-Histone (ab1791, 1:1000) from Abcam; anti-AMPKα (5831, 1:1000), anti-p-AMPKα (2535, 1:1000), anti-p-IκBα (2859, 1:1000) and anti-IκBα (4814, 1:1000) from Cell Signaling Technology; anti-NF-κB p65 (BF8005, 1:500), anti-p-NF-κB p65 (AF2006, 1:500), anti-β-actin (AF7018, 1:8000), anti-Bax (AF0120, 1:1000) and anti-Bcl-2 (AF6139, 1:500) from Affinity. An enhanced chemiluminescence solution from Bio-Rad (Irvine, CA, USA) was used to visualize immunoreactive bands. The intensity of bands was quantified using Image J 2.0.0-rc-30 Software.

### 4.9. Statistical Analysis

SPSS 19.0 software (SPSS Inc, Chicago, IL, USA) was used for the analysis of data. One-way ANOVA was used for comparisons involving over three groups. Student’s *t*-test was conducted for comparisons incorporating two groups. Statistical significance was taken at *p* < 0.05. Lowercase letters were used to represent significant differences (*p* < 0.05). Results were expressed as the mean ± SEM.

## 5. Conclusions

Rutin improved oxidative stress, ERS, apoptosis, and inflammation in H_2_O_2_-treated BMEC by activating the AMPK/NFE2L2 signaling pathway. To summarize, the findings of this research offer an experimental foundation for utilizing rutin in the treatment of mastitis or other metabolic disorders, while also contributing to the development of therapeutic strategies for mammary gland injuries in dairy cows.

## Figures and Tables

**Figure 1 ijms-26-04788-f001:**
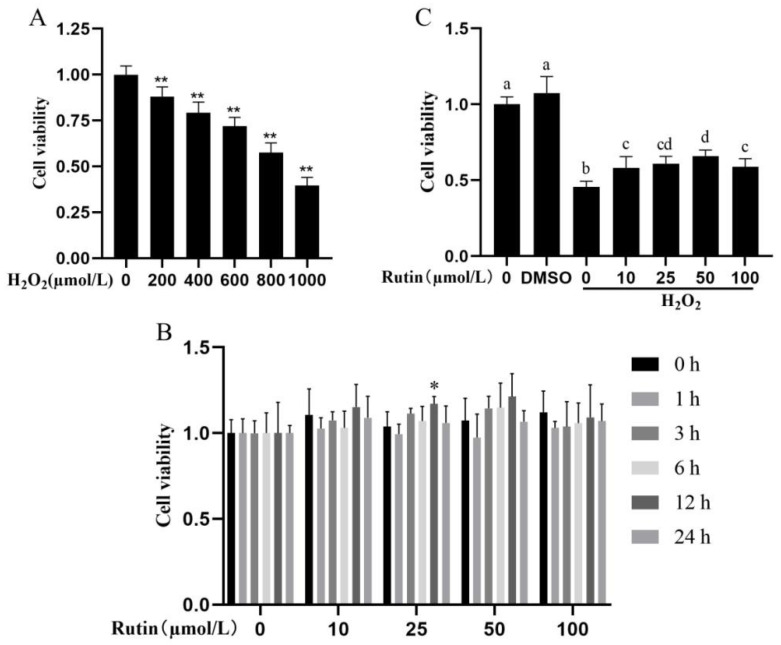
Cell viability was affected by rutin and H_2_O_2_. (**A**) BMECs were treated with H_2_O_2_ (0, 200, 400, 600, 800, and 1000 μmol/L) for 6 h. (**B**) BMECs were treated with rutin (0, 10, 25, 50, and 100 μmol/L) for 0, 1, 3, 6, 12, and 24 h. (**C**) BMECs were pretreated with rutin (0, 10, 25, 50, 100, and 200 μmol/L) for 12 h and/or H_2_O_2_ (600 μmol/L) was treated for 6 h. H_2_O_2_, hydrogen peroxide. * *p* < 0.05 and ** *p* < 0.01. Means exhibiting significant differences compared with other groups are marked with various lowercase letters (*p* < 0.05).

**Figure 2 ijms-26-04788-f002:**
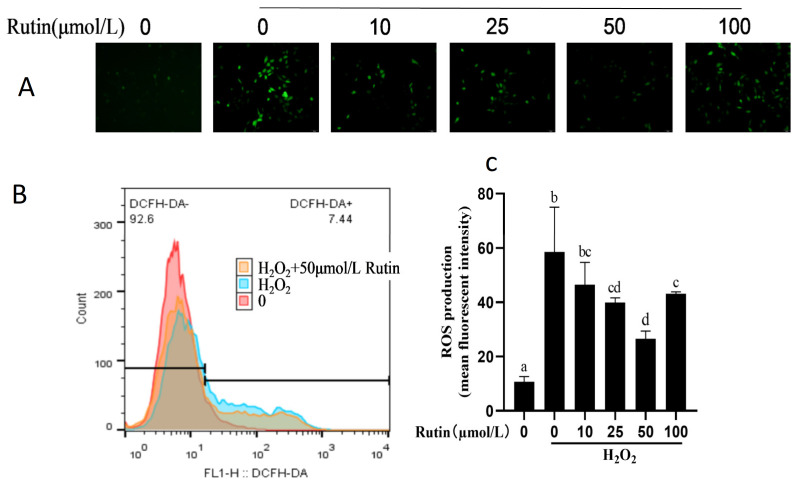
Influences on ROS in BMECs treated with H_2_O_2_ and rutin. BMEC underwent 12 h pretreatment with rutin at specified concentrations and then a 6 h challenge with H_2_O_2_ (600 μmol/L). (**A**) ROS concentrations were measured by fluorescence microscopy (200×). (**B**,**C**) ROS concentrations were measured by flow cytometer. ROS, reactive oxygen species; H_2_O_2_, hydrogen peroxide. The different lowercase letters indicated significant difference (*p* < 0.05); H_2_O_2_ indacated BMEC treated by 600 μmol/L H_2_O_2_ for 6 h.

**Figure 3 ijms-26-04788-f003:**
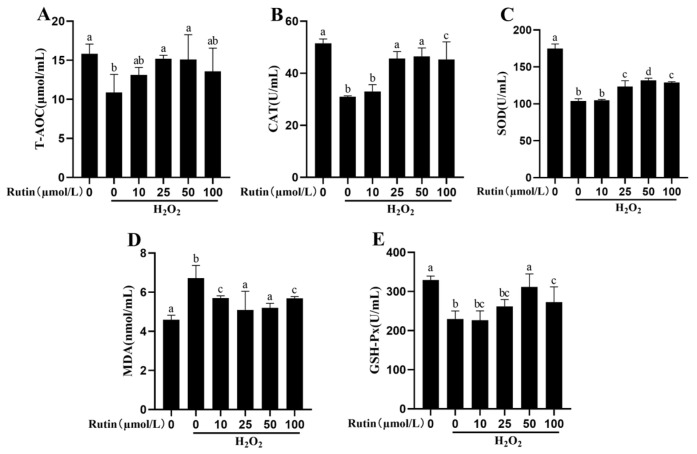
Rutin attenuated H_2_O_2_-induced oxidative stress. BMECs pretreated for 12 h with rutin at indicated concentrations together with a challenge for 6 h with H_2_O_2_ (600 μmol/L). (**A**) MDA content in BMECs. (**B**) SOD activity in BMECs. (**C**) GSH-Px activity detected from BMECs. (**D**) CAT activity in BMECs. (**E**) T-AOC content in BMECs. CAT, catalase; H_2_O_2_, hydrogen peroxide; GSH-Px, glutathione peroxidase; T-AOC, total antioxidant capacity; MDA, malondialdehyde; and SOD, superoxide dismutase. The different lowercase letters indicated significant difference (*p* < 0.05); H_2_O_2_ indacated BMEC treated by 600 μmol/L H_2_O_2_ for 6 h.

**Figure 4 ijms-26-04788-f004:**
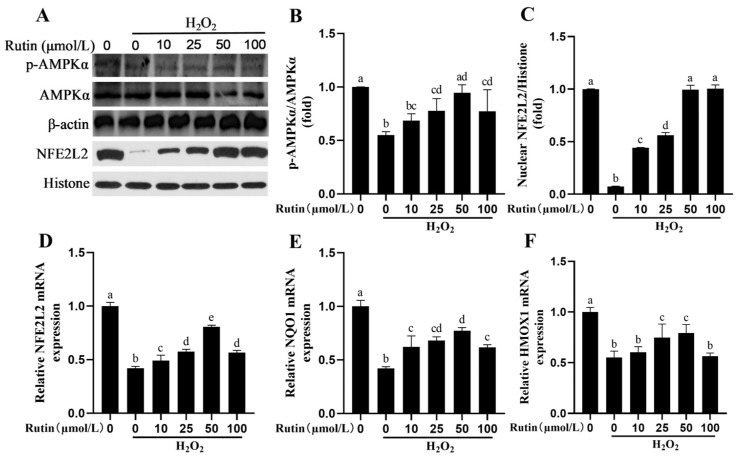
Effects of rutin pretreatment on the AMPK/NFE2L2 signaling pathway induced by H_2_O_2_ in BMECs. BMECs were pretreated with specified concentrations of rutin for 12 h, before a 6 h challenge with H_2_O_2_ (600 μmol/L). (**A**) BMEC expression of p-AMPKα and AMPKα proteins in the cytoplasm and the NEFE2L2 protein in the nucleus detected by Western blotting; (**B**) Effects of different concentrations of rutin on the ratio of p-AMPKα/AMPKα induced by H_2_O_2_ in BMECs; (**C**) Effects of rutin concentration on NFE2L2 protein expression in the nucleus induced by H_2_O_2_ in BMECs; (**D**–**F**) Varying concentrations of rutin on the relative mRNA expression of *NFE2L2*, *NQO1*, and *HMOX1* induced by H_2_O_2_ in BMECs. NQO1, NADH dehydrogenase quinone 1; AMPKα, AMP-activated protein kinase alpha subunit; NFE2L2, Nuclear factor, erythroid 2-like 2; and HMOX1, Heme oxygenase 1. The different lowercase letters indicated significant difference (*p* < 0.05); H_2_O_2_ indacated BMEC treated by 600 μmol/L H_2_O_2_ for 6 h.

**Figure 5 ijms-26-04788-f005:**
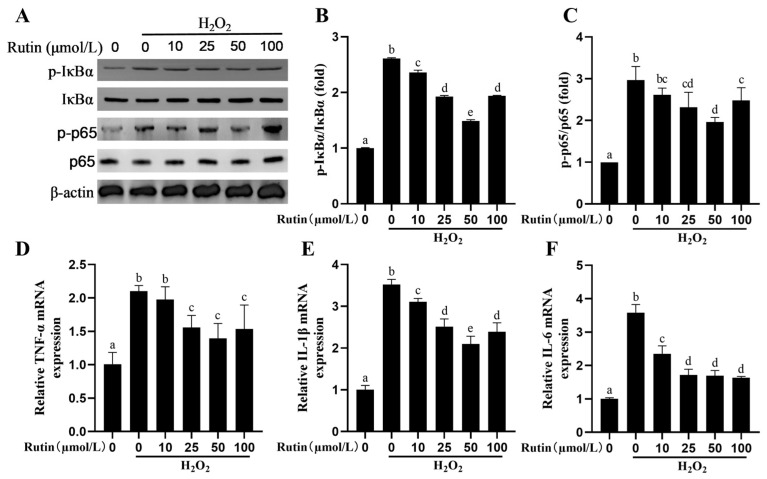
Effects of pretreating BMECs with rutin on inflammation-related indexes induced by H_2_O_2_. (**A**) Western blotting was used to determine the expression of p-IκBα, IκBα, p-p65 and p65 proteins in BMECs. (**B**,**C**) Effects of different concentrations of rutin on the p-p65/p65 plus p-IκBα/IκBα ratios induced by H_2_O_2_ in BMECs. (**D**–**F**) Effects of different concentrations of rutin on H_2_O_2_-triggered mRNA expression level of *TNF-α*, *IL-1β*, and *IL-6* at in BMECs. IL-6, Interleukin-6; IκBα, Nuclear Factor of kappa light chain gene enhancer in B-cell inhibitor, alpha; TNF-α, Tumor Necrosis Factor-alpha; p65, RelA-associated inhibitor of NF-kappaB; and IL-1β, Interleukin-1 beta. The different lowercase letters indicated significant difference (*p* < 0.05); H_2_O_2_ indacated BMEC treated by 600 μmol/L H_2_O_2_ for 6 h.

**Figure 6 ijms-26-04788-f006:**
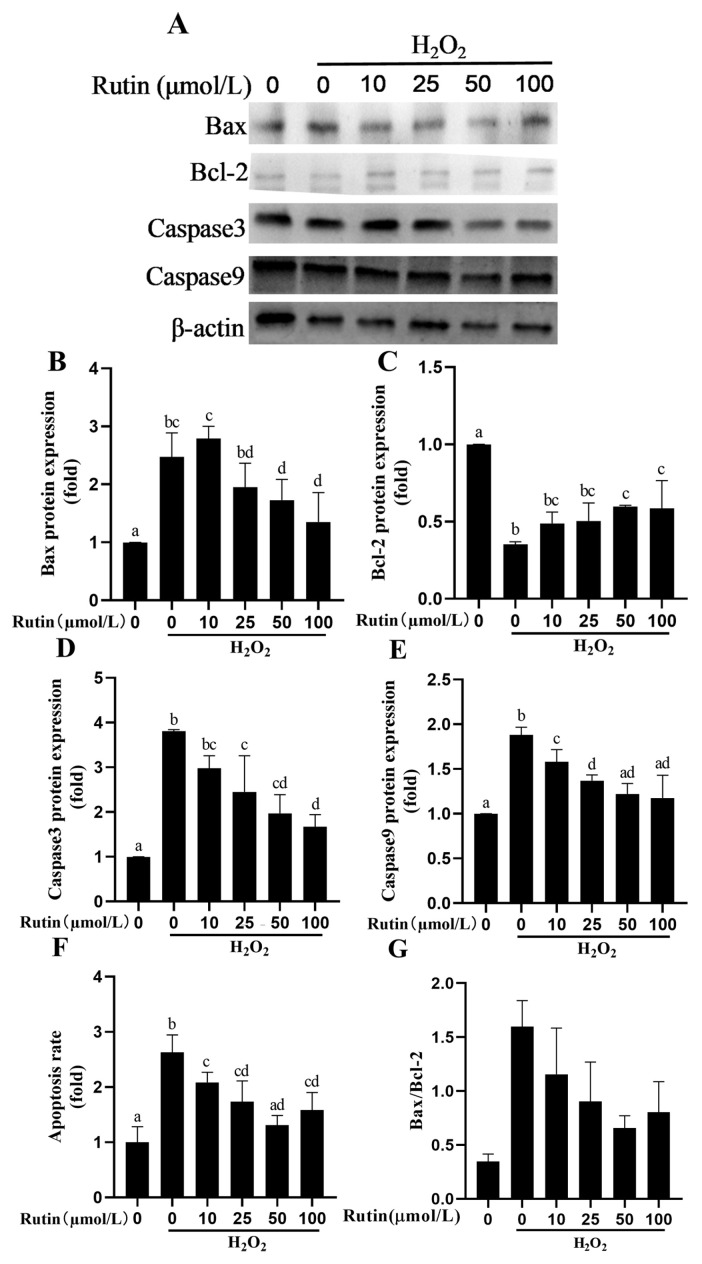
Effects of rutin pretreatment on apoptosis-related indexes induced by H_2_O_2_ in BMECs. (**A**) Bax, Bcl-2, Caspase3, and Caspase9 proteins were detected in BMECs by Western blotting; (**B**–**E**) Effect of different rutin concentrations on protein expression of Bax, Bcl-2, Caspase3 and Caspase9 induced by H_2_O_2_ in BMECs; (**F**) Effects of different rutin concentrations on the apoptosis ratio induced by H_2_O_2_ in BMECs. Bcl-2, B-cell lymphoma 2; and Bax, Bcl2-associated X protein. (**G**) The Bax/Bcl-2 ratio in H_2_O_2_ and rutin treated BMECs. The different lowercase letters indicated significant difference (*p* < 0.05); H_2_O_2_ indacated BMEC treated by 600 μmol/L H_2_O_2_ for 6 h.

**Figure 7 ijms-26-04788-f007:**
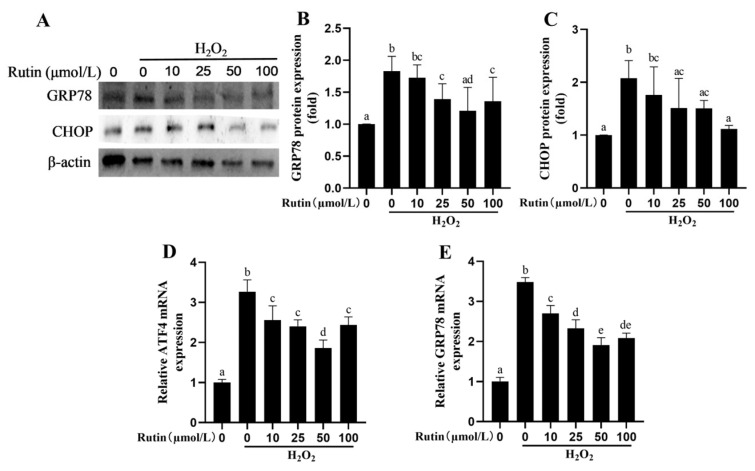
Effects of rutin pretreatment on ERS-related indexes induced by H_2_O_2_ in BMECs. (**A**) Western blotting was used to measure the expression of GRP78 and CHOP proteins in BMECs. (**B**,**C**) Effects of different rutin concentrations on H_2_O_2_-mediated GRP78 and CHOP protein expressions in BMECs. (**D**,**E**) Effects of different rutin concentrations on H_2_O_2_-induced *GRP78* and *ATF4* mRNA expression in BMECs. GRP78, Glucose-regulated protein 78; ATF4, Activating Transcription Factor 4; and CHOP, C/EBP homologous protein. The different lowercase letters indicated significant difference (*p* < 0.05); H_2_O_2_ indacated BMEC treated by 600 μmol/L H_2_O_2_ for 6 h.

**Figure 8 ijms-26-04788-f008:**
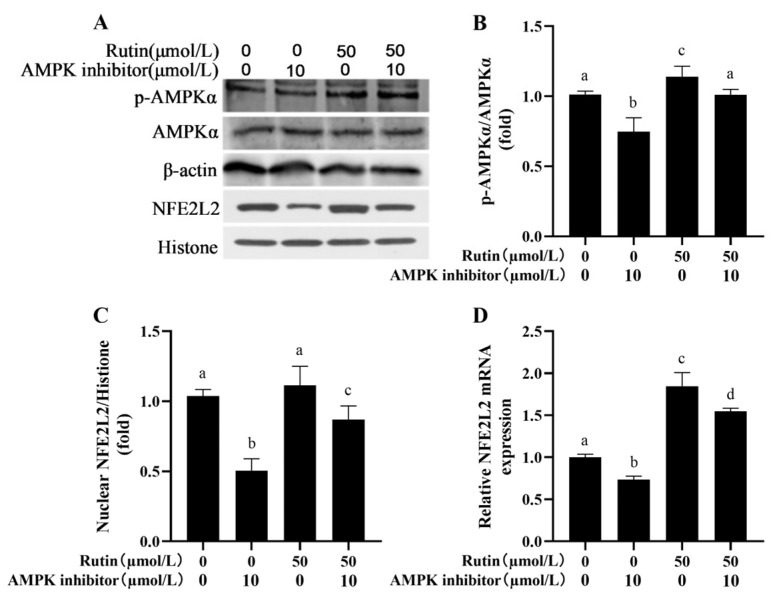
Rutin upregulated the expression of NFE2L2 in BMECs through AMPK. (**A**) Expression of p-AMPKα and AMPKα proteins in the cytoplasm and the expression of NFE2L2 protein in the nucleus of BMECs were detected by Western blotting. (**B**) Effects of AMPK inhibitor on the ratio of p-AMPKα/AMPKα in rutin-treated BMECs. (**C**) AMPK inhibitor influencing NFE2L2 protein expression in the nucleus of BMECs treated with rutin. (**D**) Effects of AMPK inhibitor on *NFE2L2* mRNA expression in rutin-treated BMECs. AMPKα, AMP-activated protein kinase alpha subunit; and NFE2L2, Nuclear factor erythroid 2-like 2. The different lowercase letters indicated significant difference (*p* < 0.05); H_2_O_2_ indacated BMEC treated by 600 μmol/L H_2_O_2_ for 6 h.

**Figure 9 ijms-26-04788-f009:**
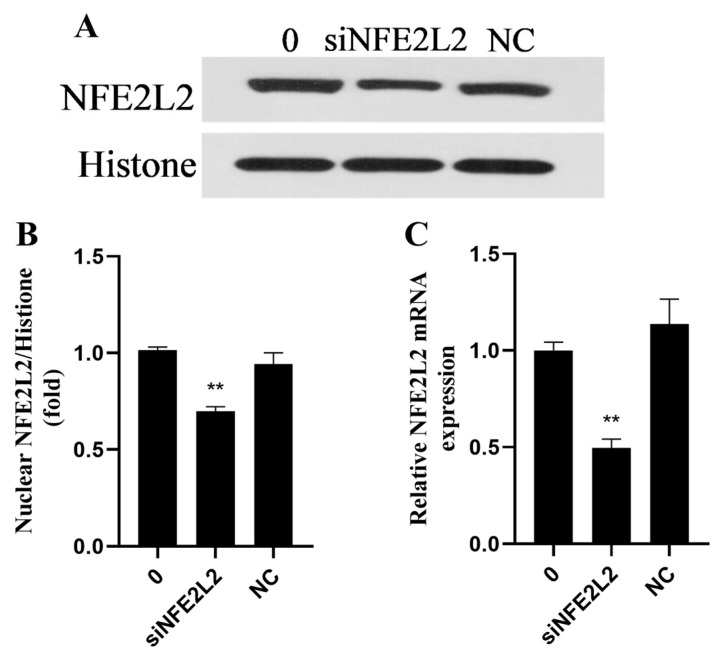
The inhibitory effect of siNFE2L2 on NFE2L2 in BMECs. (**A**) Expression of NFE2L2 protein in the nucleus of BMECs was detected by Western blotting. (**B**) Effects of positive and negative siRNA on the expression of the NFE2L2 protein in the nucleus of BMECs treated with rutin. (**C**) Effects of positive and negative siRNA on the relative mRNA expression of *NFE2L2* in BMECs treated with rutin. NC, Negative control group; NFE2L2, Nuclear factor, erythroid 2-like 2; and siNFE2L2, NFE2L2 siRNA. ** *p* < 0.01.

**Figure 10 ijms-26-04788-f010:**
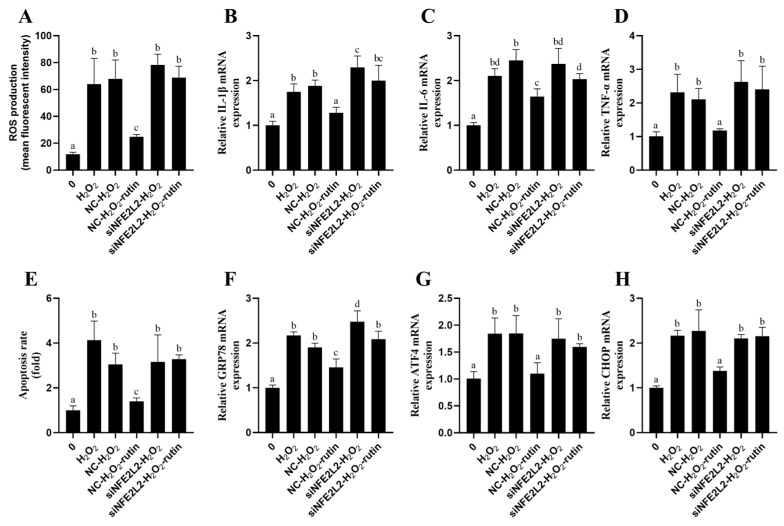
Rutin alleviates oxidative damage induced by H_2_O_2_ in BMECs through NFE2L2. (**A**) After NFE2L2 siRNA knockdown, the effects of rutin on H_2_O_2_-induced ROS content in BMECs. (**B**–**D**) After NFE2L2 siRNA knockdown, efficacy of rutin in affecting the H_2_O_2_-mediated relative mRNA expression of inflammation-related factors (*TNF-α*, *IL-6* and *IL-1β*) in BMECs. (**E**) After NFE2L2 siRNA knockdown, influence of rutin on H_2_O_2_-induced BMEC apoptosis rate. (**F**–**H**) After NFE2L2 knockdown with siRNA, impact of rutin on relative H_2_O_2_-triggered mRNA expression of ERS-related factors (*GRP78*, *ATF4* and *CHOP*) in BMECs. ATF4, Activating Transcription Factor 4; GRP78, Glucose-regulated protein 78; IL-6, Interleukin-6; CHOP, C/EBP homologous protein; TNF-α, Tumor Necrosis Factor-alpha; and IL-1β, Interleukin-1 beta. The different lowercase letters indicated significant difference (*p* < 0.05).

**Table 1 ijms-26-04788-t001:** Primer sequences of the genes.

Genes	Primer Sequences (5′-3′)		Length (bp)
*β-actin*	Forward	CCTGGAGAAGAGCTACGAG	132
	Reverse	AAGGTAGTTTCGTGAATGCC	
*TNF-α*	Forward	CAAGCCTCAAGTAACAAGCC	123
	Reverse	GTTGTCTTCCAGCTTCACAC	
*IL-6*	Forward	AAGGAGACACTGGCAGAAAA	91
	Reverse	AGCAAATCGCCTGATTGAAC	
*IL-1β*	Forward	GGATCCTATTCTCTCCAGCC	82
	Reverse	TTTCGTTGATGGGTTCAGGT	
*GRP-78*	Forward	GTGCGTTTGAGAGCTCAGTA	90
	Reverse	GTCAGCAGTCAGCTATAGGG	
*ATF4*	Forward	GGCTCTCCAAATAAGAGCCT	106
	Reverse	CCTTGACTTTTGCTGCTACC	
*NFE2L2*	Forward	TCAGCATGATGGACTTGGAG	114
	Reverse	ATACCTCTCGACTTACCCCA	
*HMOX1*	Forward	GTCCTCACACTCAGCTTTCT	192
	Reverse	AGAGAGGGATACAGGGAGAC	
*NQO-1*	Forward	ACTTTCAGTATCCTGCCGAG	193
	Reverse	TACTTGTAAGCGAACTCCCC	
*CHOP*	Forward	GGTCAGAGGCTTGAGTCTAA	91
	Reverse	CTCAGGTTCCGGCTTTGATT	

## Data Availability

Study data are available to the corresponding author by email.

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
