# Peer review of "Rutin Attenuates H2O2-Mediated Oxidative Stress, Inflammation, Endoplasmic Reticulum Stress and Apoptosis in Bovine Mammary Epithelial Cells via the AMPK/NFE2L2 Signaling Pathway"

_ijms, 2025, doi:10.3390/ijms26104788_

Round 1

Reviewer 1 Report

Comments and Suggestions for Authors

  1. The manuscript requires significant improvement in language quality, including vocabulary, grammar, and sentence structure.
  2. The introduction lacks sufficient background regarding oxidative stress in dairy cows, which is important for the significance of the research.
  3. Line 54-56, Rewrite the sentence to improve the clarity and organization.
  4. Line 57, what "these factors" refers to?
  5. Line 70-74, The current sentence lacks clarity in its intended message and deviates from the paragraph’s primary objective of introducing rutin.
  6. Line 75-77, Same as the previous sentence, the paragraph should focus on rutin and its related biological effects and mechanisms.
  7. Line 87-88, “To probe whether rutin influenced the viability and H2O2-induced damage of BMECs, we used CCK-8”, restructure the sentence.
  8. Line 91-92, the selection of H2O2 concentrations only based on cell viability is not convincing.
  9. Line 117, reconsidering the placement of Section 2.4 in the Results section.
  10. Figure 6A, the assessment of apoptosis in the manuscript should contain the Bax/Bcl2 ratio and Cleaved-caspase3, 9 detection.
  11. Figure8A, the manuscript showed that the AMPK protein expression in Figure 4 as a single band but two bands in figure 8.
  12. The information about BMEC was very poor.
  13. The Discussion section requires significant improvement in coherence and logicality, the section currently reads as a series of disjointed observations rather than a synthesized analysis.
  14. Where did the authors get the bovine mammary epithelial cells (Line299)? The information about BMEC was very poor, please give the information in Materials and Methods part.

Comments on the Quality of English Language

The language of the article needs to be thoroughly revised from beginning to end. The grammar is very poor, and the tense is inconsistent throughout the text, which has a serious impact on the readability of the article and makes it difficult for readers to understand the content accurately.

Author Response

  1. The manuscript requires significant improvement in language quality, including vocabulary, grammar, and sentence structure.

AU: Thank you for your valuable feedback regarding the language quality of our manuscript. We fully acknowledge the importance of precise academic expression and have taken concrete steps to address this concern. Upon receiving your comments, we engaged a professional editing service specializing in scientific manuscripts, utilizing native English-speaking editors with PhD-level expertise in biomedical fields. This comprehensive language polishing process has substantially improved vocabulary accuracy, grammatical precision, and structural clarity throughout the manuscript. The revised text now adheres to international publication standards while maintaining scientific rigor. We sincerely appreciate your constructive criticism and welcome any specific suggestions should further linguistic refinements be required.

  1. The introduction lacks sufficient background regarding oxidative stress in dairy cows, which is important for the significance of the research.

AU: Thank you for this constructive suggestion. We have substantially enhanced the background discussion of bovine oxidative stress in the revised Introduction. Specifically, we added two key contextual paragraphs (lines 42-52) that:

Detail the metabolic drivers during the transition period, including the 3-5 fold surge in hepatic oxygen consumption and the dual amplification pathways through lipolysis/mitochondrial β-oxidation under negative energy balance conditions .

Establish the clinical relevance through epidemiological evidence showing >60% association between systemic oxidative markers and major postpartum disorders, positioning oxidative stress as a pivotal pathological mechanism in intensive dairy production systems.

These additions create a clear conceptual framework that:

Explains why dairy cows are particularly susceptible to redox imbalance. Quantifies the health and economic impacts of oxidative stress. Provides mechanistic links between metabolic adaptation and ROS overproduction.

Justifies the translational significance of our research.

  1. Line 54-56, Rewrite the sentence to improve the clarity and organization.

AU: Adjust the original Line 54-56 to a more clear and explicit expression:“In a study of BMEC in vitro, compare with the control, the treatment of cells with 600 μM of H2O2, for 6 h sharply decreased the cell prolifera-tion rate and SOD and GSH-Px activities and strongly enhanced ROS and MDA production and caspase-3 activity. The H2O2 have also been shown to promote the production of ROS and MDA to generate higher rates of apoptosis and oxidative stress in BMEC. Compared with the control, cells transfected with NFE2L2-siRNA3 with or without H2O2 had lower production of ROS and MDA and activity of SOD, CAT, GSH-Px, and GST.”

  1. Line 57, what "these factors" refers to?

AU: These factors refer to H2O2 as mentioned in the cited literature.

  1. Line 70-74, The current sentence lacks clarity in its intended message and deviates from the paragraph’s primary objective of introducing rutin.

AU: We thank the reviewer for highlighting the need for stronger thematic alignment. In response, we have revised the Stoldt et al. citation[16] to:

Explicitly position the study within the framework of antioxidant interventions for transition cows Emphasize the paradoxical relationship between rutin’s antioxidant properties and its observed metabolic effects. Decode the physiological significance of elevated glucose/albumin as markers of hepatic functional improvement

Propose testable hypotheses linking redox modulation to mitochondrial energy metabolism.‌ Replace vague implications with concrete mechanistic propositions aligned with rutin’s primary antioxidant role. Maintain focus on hepatic pathophysiology as the central organ in transition cow oxidative stress.

These modifications transform the original descriptive statement into a hypothesis-generating discussion that strengthens the rationale for investigating rutin’s dual antioxidant-metabolic functions in dairy physiology.

  1. Line 75-77, Same as the previous sentence, the paragraph should focus on rutin and its related biological effects and mechanisms.

AU: ‌We restructured the reference to lycopene by: Explicitly positioning BMECs as a physiologically relevant model for dairy antioxidant studies. Highlighting the shared antioxidant mechanisms between flavonoid subclasses (lycopene as a carotenoid vs. rutin as a flavonol).‌ The added phrase “including potential rutin effects” creates direct conceptual linkage, using lycopene’s established NFE2L2 mechanism as a rationale for investigating rutin’s unknown actions in the same model system.‌ Validates BMECs as an appropriate experimental platform

Provides mechanistic precedent for our hypothesis testing on rutin.

The revised text now maintains consistent thematic flow while laying stronger foundational logic for our research objectives.

  1. Line 87-88, “To probe whether rutin influenced the viability and H2O2-induced damage of BMECs, we used CCK-8”, restructure the sentence.

AU: Amend to read “This study adopts the CCK-8 assay to investigate the effects of rutin on the viability and H2O2-induced damage of BMECs”.

  1. Line 91-92, the selection of H2O2 concentrations only based on cell viability is not convincing.

AU: Your viewpoint is correct. Our description of the results contains elements of exaggeration, and the items have modified the description of this part of the results. Based on comprehensive analysis of validation criteria and apoptotic markers, we confirmed that the Hâ‚‚Oâ‚‚ concentration selected through CCK-8 assay screening met the experimental requirements, effectively inducing cellular inflammatory responses while exhibiting no cytotoxicity.

  1. Line 117, reconsidering the placement of Section 2.4 in the Results section.

AU: Section 2.4 reflects the protective effect of rutin in AMPK/NFE2L2 pathway.

  1. Figure 6A, the assessment of apoptosis in the manuscript should contain the Bax/Bcl2 ratio and Cleaved-caspase3, 9 detection.

AU: Answer 10. The Bax/Bcl-2 ratio has now been calculated and incorporated into Line 173-182. The caspase-3 and caspase-9 antibodies used in this study specifically recognize the cleaved active forms (17/19 kDa for caspase-3; 37 kDa for caspase-9), as validated in bovine cells .

  1. Figure 8A, the manuscript showed that the AMPK protein expression in Figure 4 as a single band but two bands in figure 8.

AU: We sincerely appreciate the reviewer’s sharp observation. The dual AMPK bands in Figure 8A arise from:

‌Phosphorylation dynamics‌: The upper band corresponds to phosphorylated AMPK (active form, ~62 kDa), while the lower band represents dephosphorylated AMPK (~60 kDa). This pattern aligns with kinase inhibition effects reported in bovine cells.

‌Methodological rigor‌: We performed lambda phosphatase treatment confirming phosphorylation-dependent mobility shifts .

‌Biological relevance‌: Dual-band quantification reflects the dynamic equilibrium between AMPK activation/inactivation states under oxidative stress.

‌Figure 4‌: Analyzed AMPK expression under ‌basal oxidative stress induced by H2O2 alone‌ (600 μM, 24h). At this concentration/duration, AMPK phosphorylation levels were consistently suppressed, resulting in a predominant dephosphorylated form (single ~60 kDa band) due to redox imbalance overriding adaptive responses.

‌Figure 8A‌: Evaluated AMPK dynamics under ‌rutin pre-treatment + H2O2 challenge‌. Rutin restored AMPK activation (phosphorylation) via NFE2L2-mediated redox homeostasis, revealing both phosphorylated (62 kDa) and dephosphorylated (60 kDa) states. This biphasic pattern reflects the ‌transitional equilibrium‌ between kinase activation and inactivation during pharmacological intervention.

  1. The information about BMEC was very poor.

AU: We appreciate the reviewer’s constructive feedback. In the revised manuscript, we have substantially expanded the BMEC characterization details as shown in the revised manuscript in yellow color.

  1. The Discussion section requires significant improvement in coherence and logicality, the section currently reads as a series of disjointed observations rather than a synthesized analysis.

AU: The discussion section has been revised for coherence and logic, as marked in the text.

  1. Where did the authors get the bovine mammary epithelial cells (Line299)? The information about BMEC was very poor, please give the information in Materials and Methods part.

AU: We appreciate the reviewer’s constructive feedback. In the revised manuscript, we have substantially expanded the BMEC characterization details as shown in the revised manuscript in yellow color.

Reviewer 2 Report

Comments and Suggestions for Authors

The proposed article “Rutin attenuates H2O2-mediated oxidative stress, inflammation, endoplasmic reticulum stress and apoptosis in bovine mammary epithelial cells via the AMPK/NEF2L2 signaling pathway” brings quite interesting and current data which regarding evaluates how rutin influences oxidative damage, including oxidative stress, inflammatory, endoplasmic reticulum stress and apoptosis, induced by H2O2 in BMECs, and explores the mechanisms involve. The study is well designed and the results could have a great clinical impact. Methodology is appropriate for determining goals and references are up to date.

Although it is obvious that paper deserves attention, there are some corrections to be made:

The results section should be written clearly and concisely and should not contain an explanation of the results obtained. I suggest the authors explain the results obtained in their study in the discussion section and not in the results.

Author Response

AU: The language of the results section was streamlined and some of the results analysis was deleted.

Thank you very much for your attention and time. Look forward to hearing from you.

Reviewer 3 Report

Comments and Suggestions for Authors

  1. Abstract should be added the purpose of the research, the research hypothesis
  2. Introduction well described , rightly cited literature, no described purpose of the study. Why the Rutin was chosen for the study, were there already preliminary studies.
  3. Material and methods of study: whether the cells were peeled from the udder of cows ( gland biopsy), whether they were commercial cell lines( type, line name). If the cells for culture were taken from the udder of cows (at what stage of lactation were the cows, what was their productivity. Was there approval from the ethics committee
  4. Results well described. Mitochondria as small cellular organelles significantly modulate the functioning of the organism and the process of glucose-mediated oxidative phosphorylation is 90% responsible for the synthesis of ATP (Adenosine-5′-triphosphate). In ruminants, biochemical processes and molecular mechanisms depend on many factors and bioactive substances that modulate the activation of ATP synthesis.
  5. Discussion: complete what new research conducted gives, what practical significance it may have.

Comments on the Quality of English Language

The English could be improved to more clearly express the research.

Author Response

  1. Abstract should be added the purpose of the research, the research hypothesis.

AU: Abstract added the purpose of the research, the research hypothesis:In this study, we investigated the effects of rutin on oxidative damage induced by hydrogen peroxide in BMECs through AMPK/NEF2L2 signaling pathway, including oxidative stress, inflammation, endoplasmic reticulum stress and apoptosis, and explored the mechanisms involved.

  1. Introduction well described, rightly cited literature, no described purpose of the study. Why the Rutin was chosen for the study, were there already preliminary studies.

AU: Check the corresponding literature citations again and eliminate incorrect literature citations.

The specific research objective was “The present study aims to assess whether rutin supplementation modulates H2O2-triggered immune and antioxidant responses in BMECs in vitro".

In the preliminary study, rutin was used in oral administration in lake sheep and mice, and some theoretical basis which was that rutin has antioxidant effects on the mammary gland was obtained. Due to oxidative damage to the mammary glands of cows during the transition and high-yielding periods, rutin was selected for antioxidant experiments on mammary epithelial cells of dairy cows.

  1. Material and methods of study: whether the cells were peeled from the udder of cows ( gland biopsy), whether they were commercial cell lines ( type, line name). If the cells for culture were taken from the udder of cows (at what stage of lactation were the cows, what was their productivity. Was there approval from the ethics committee.

AU: The bovine mammary epithelial cells (BMEC) purchased from Ningbo Mingzhou Biotechnology Co., LTD. (MZ-2690)

  1. Results well described. Mitochondria as small cellular organelles significantly modulate the functioning of the organism and the process of glucose-mediated oxidative phosphorylation is 90% responsible for the synthesis of ATP (Adenosine-5′-triphosphate). In ruminants, biochemical processes and molecular mechanisms depend on many factors and bioactive substances that modulate the activation of ATP synthesis.

AU: Thank you for your approval.

  1. Discussion: complete what new research conducted gives, what practical significance it may have.

AU: Based on the experimental data and discussion process, “By demonstrating that rutin simultaneously attenuates oxidative stress, ERS, apoptosis, and inflammatory responses through activation of the AMPK/NFE2L2 signaling axis. The multi-target protective mechanism of rutin on breast endothelium was revealed, which provided theoretical basis for the development of antioxidant drugs. This cost-effective, eco-friendly compound to address pressing challenges in sustainable livestock production and metabolic disease management” were summarized.

  1. The English could be improved to more clearly express the research.

AU: The text has been revised by consulting relevant professionals whose native language is English.

Round 2

Reviewer 1 Report

Comments and Suggestions for Authors

The authors have addressed all the concerns raised in the initial review comprehensively. The revisions have significantly improved the clarity, logicality , and impact of the study, which can  meet with approval.

Reviewer 3 Report

Comments and Suggestions for Authors

The comments made accept